# Effect of Progressive Head Extension Swallowing Exercise on Lingual Strength in the Elderly: A Randomized Controlled Trial

**DOI:** 10.3390/jcm10153419

**Published:** 2021-07-31

**Authors:** Jin-Woo Park, Chi-Hoon Oh, Bo-Un Choi, Ho-Jin Hong, Joong-Hee Park, Tae-Yeon Kim, Yong-Jin Cho

**Affiliations:** Department of Physical Medicine and Rehabilitation, Dongguk University Ilsan Hospital, Goyang-si 10326, Gyeonggi-do, Korea; chejuoh@hanmail.net (C.-H.O.); moongirl33@naver.com (B.-U.C.); frischen@naver.com (H.-J.H.); s65271@hanmail.net (J.-H.P.); tinaccjj@naver.com (T.-Y.K.); pigboom@hanmail.net (Y.-J.C.)

**Keywords:** deglutition disorders, tongue, exercise, deglutition, ageing

## Abstract

Lingual strengthening training can improve the swallowing function in older adults, but the optimal method is unclear. We investigated the effects of a new progressive resistance exercise in the elderly by comparing with a conventional isometric tongue strengthening exercise. Twenty-nine participants were divided into two groups randomly. One group performed forceful swallow of 2 mL of water every 10 s for 20 min, and a total of 120 swallowing tasks per session at 80% angle of maximum head extension. The other group performed five repetitions in 24 sets with a 30 s rest, and the target level was settled at 80% of one repetition maximum using the Iowa Oral Performance Instrument (IOPI). A total of 12 sessions were carried out by both groups over a 4-week period. Blinded measurements (for maximum lingual isometric pressure and peak pressure during swallowing) were obtained using IOPI before exercise and at four weeks in both groups. After four weeks, both groups showed a significant improvement in lingual strength involving both isometric and swallowing tasks. However, there was no significant difference between the groups in strength increase involving both tasks. Regardless of the manner, tongue-strengthening exercises substantially improved lingual pressure in the elderly with equal effect.

## 1. Introduction

Presbyphagia means characteristic alteration in the deglutition mechanism of healthy older adults [1]. Aging worsens motor swallowing mechanism, which, in turn, leads to weakness in tongue muscle [2]. It is significant that the tongue is the main source of propelling oropharyngeal swallowing [3], and abnormal tongue strength and coordination can decrease the safety and efficiency of swallowing [4,5].

Fortunately, tongue exercises can increase tongue strength and improve swallowing ability in older people. In this way, exercise using an air bulb or pushing against hard palate as a resistive isometric exercise can improve tongue strength and swallowing function [6,7]. Real swallowing exercise can also improve tongue strength in the elderly [8]. However, the method that is the best for increasing tongue strength is currently unclear.

We know that the training method based on the basic principle of exercise is the best [9]. Training specificity means that improvement in performance is most dramatic when movements closely coincide with the exercise. When applied to the tongue, the tongue strength is improved during swallowing. According to the overload principle, exercise resistance should be gradually increased as the individual capabilities improve throughout the training. Exercises using an air bulb or tongue depressor [6,10,11] are resistive isometric exercises and appropriate for the overload principle but are not based on training specificity. Actual swallowing exercises such as effortful swallow [12] are based on training specificity, but they do not adhere to the overload principle because the exercise intensity cannot be adjusted.

However, head extension swallowing exercises can increase lingual swallowing pressure and endurance in an older adult population [13]. Even though this exercise is based on the work of a single research group involving a limited number of people which has yet to be replicated elsewhere, it can be easily performed anytime and anywhere without the need for additional equipment, especially given the benefits of resistance exercise. We thought that it might conform to training specificity and overload principle, and effectively improve tongue strength. We modified this exercise by adjusting the angle of head extension in order to control and increase the intensity of the exercise (progressive resistance exercise). We hypothesized that this new exercise is effective in increasing tongue strength in older adults, and that the exercise is superior to the lingual elevation exercise. Therefore, in this study, we analyzed the effects of a new progressive resistance exercise for performance by older adults, and we compared the results with conventional isometric tongue-strengthening exercises.

## 2. Materials and Methods

### 2.1. Participants

Thirty-five healthy older volunteers were eligible for this study, which was conducted from August 2019 to February 2020. The inclusion criteria were: (1) healthy older people aged above 65 years without dysphagia, and (2) sufficient cognitive function to perform tongue-strengthening exercises (mini-mental status exam ≥ 26). Thus, the exclusion criteria were: (1) history of odynophagia or dysphagia, (2) drugs that influence swallowing, and (3) history of cervical spine disease that prohibits head extension. Before attending this study, all of the participants were examined by a doctor. This study adheres to CONSORT guidelines, and the Institutional Review Board approved this study. Informed consent was obtained from each subject. Twenty-nine volunteers participated in this study, and 26 of the 29 participants who completed the 12 sessions of the exercise were included in this analysis (Figure 1). Three of the 29 participants dropped out after performing the exercise 2 to 3 times because they either had no time to visit the hospital or their place of residence was located too far from the hospital. The mean age of the study group was 72.9 ± 6.4 years, and the study included 5 males and 21 females. The general characteristics of these volunteers are shown in Table 1.

### 2.2. Experimental Protocol 

This study was designed as a randomized, controlled study and was scheduled for a total of 4 weeks. The study participants were randomly allocated to two groups with a 1:1 ratio: tongue progressive resistance exercise group (G1) or tongue isometric exercise group (G2) using a randomization computer program. The assessor and statistical analyst were unaware of the group assignment. Before strengthening training, we measured the baseline data including maximum lingual isometric pressure and peak pressure during swallowing using Iowa Oral Performance Instrument (IOPI) (model 2.1; IOPI Medical LLC, Carnation, WA, USA), which is a handheld tool for measuring the pressure on a small air-filled bulb [14]. Each strengthening program was then administered to the participants over a course of 4 weeks, followed by reassessment of strength to evaluate the training effects of the tongue-strengthening exercise.

### 2.3. Tongue Strengthening Training

The G1 group performed an effortful swallow of 2 mL of water every 10 s for 20 min with a total of 120 swallowing tasks per session at 80% angle of maximum head extension (MHE). One session consisted of two 10 min period exercises with a 5 min period rest between exercises to avoid muscle fatigue. All participants received instruction to maintain the same posture by staring at one point during the swallowing attempts. The point was determined to ensure that the participants looked comfortable by staring at the grid on the wall 1 m away while maintaining the determined head extension angle. Next, the G2 group did an exercise, which consisted of five repetitions, 24 sets, 30 s rest between sets and a total of 120 lingual pressing tasks per session, with the target level set at 80% of one repetition maximum (RM) using an IOPI. Participants hold the bulb for 3 s based on the light-emitting diode (LED). MHE and one RM were repeatedly measured every week and the exercise levels were readjusted. Three sessions were performed by both groups each week over a 4-week duration (total 12 sessions). All exercises were carried out in the University Hospital under supervision. 

### 2.4. Head Extension Measurements

Each participant in the G1 group sat on a chair ensuring that the thoracic vertebrae were in constant contact with the back of the chair, and the lumbar vertebrae filled the gap between the seat and the back. The participant’s feet were placed flat on the floor and arms were placed freely at their sides. Next, the inclinometer (Baseline^®^ Bubble Inclinometer, FEI, White Plains, NY, USA) was mounted over the participant’s vertex of the head. Next, the tester instructed each participant to extend his or her head until they could not swallow volitionally, and then measured the MHE angle using the inclinometer (Figure 2).

### 2.5. Tongue Strength Measurements

In the study, the blinded lingual pressures were measured using IOPI with participants seated comfortably in an upright position during two different tasks: (1) maximum isometric pressure and (2) peak pressure during saliva swallowing [15]. The bulb was positioned at 10 mm anterior to the most posterior circumvallate and pressures (expressed in kPa) were displayed on a liquid crystal display (LCD) panel on the device. For the isometric task, volunteers received instruction to press the bulb against the “roof of the mouth” with the tongue as “hard as possible.” For the swallowing task, the participants were instructed to swallow saliva as they would normally with the bulb in place. Three trials to generate maximal pressures were attempted and the highest pressure was used to measure the tongue strength.

### 2.6. Statistical Analysis

The statistical analysis was carried out using SPSS version 12.0 (SPSS, Inc., Chicago, IL, USA). For determining the sample size, the predicted difference (d) of IOPI was set to 5 and the standard deviation S was set to 5. An alpha error of 0.05 and a beta error of 0.2 were calculated to arrive at a total of 32 subjects. Group comparisons of baseline demographics were performed using Student’s *t*-test for continuous variables and χ^2^ test for categorical variables to test imbalance between groups. Likewise, the paired *t*-test was used for comparison between paired variables (pre- and post-training in groups). Finally, the comparison of the absolute increase in strength between groups was performed with Student’s *t*-test. The significance level was set at *p* < 0.025 to consider alpha-level adjustments for multiple comparisons.

## 3. Results

The mean baseline maximum head extension angle in G1 was 39.6 ± 9.9 (25–55) degrees, which significantly increased to 57.7 ± 7.8 (40–70) degrees after 4 weeks. The increase in maximum head extension angle was positively correlated with the increase in tongue strength in the G1 group (Spearman’s Rho, r = 0.651, *p* = 0.016)

The average baseline maximum isometric pressures (average ± standard deviation) of G1 and G2 were 40.5 ± 9.2 kPa and 43.5 ± 10.4 kPa, respectively, showing no significant differences between groups (*p* = 0.455). The average baseline peak pressures during swallowing of G1 and G2 were 26.1 ± 12.4 kPa and 31.3 ± 12.6 kPa, respectively, and also there was no significant difference between the groups (*p* = 0.297). After four weeks of exercise, the tongue strength in both isometric and swallowing tasks was increased significantly in both groups (G1, *p* < 0.001, Cohen’s d = 2.222 and G2, *p* < 0.001, Cohen’s d = 1.469 for isometric pressure; G1, *p* = 0.001, Cohen’s d = 0.882 and G2, *p* = 0.003, Cohen’s d = 0.763 for pressure during swallowing) (Figure 3). However, no significant difference in strength increment in both tasks was detected between the groups (G1, 17.6 ± 7.5 kPa and G2, 14.0 ± 7.9 kPa, *p* = 0.244 for isometric pressure; G1, 11.9 ± 10.3 kPa and G2, 10.2 ± 10.1 kPa, *p* = 0.662 for pressure during swallowing) (Figure 4).

## 4. Discussion

Four weeks of progressive head extension swallowing exercise improved tongue strength in older volunteers. However, this method was not superior to conventional isometric strengthening exercise. Likewise, the head extension swallowing exercise strengthens the tongue and suprahyoid muscles. It was originally a compensatory method administered to inpatients with head and neck cancer who generally present with problems associated with oral food intake [16]. However, the use of head extension as a resistance mechanism to strengthen the tongue was applicable to young and old alike [13,17]. We modified this exercise by additionally increasing the angle of head extension to control the intensity of exercise. Progressive head extension swallow training that meets training specificity criteria and overload principle is expected to be the most effective method to increase lingual strength.

However, lingual strengthening training does not follow standard exercise principles. In fact, the unique physiology of the lingual musculature may defy many types of exercise principles [18]. The tongue is a muscular hydrostat, which generates force via contraction of muscle fibers to generate hydraulic pressure within a limited area. However, the muscles of the human tongue are unique in that they are attached to only a single static support (mandible or styloid process), or to a floating support (hyoid bone). The tongue is a cylindrical structure with a constant volume that adjusts its shape and size by co-activating many of its muscular components. The implication in this case is that because the muscles cannot contract by attaching to a bony support, as in the arm or leg, the hydrostatic pull on the muscles results in a net productive movement. In contrast, skeletal muscles usually contract with joints to create force, and most of the theory underlying exercise physiology is based on skeletal muscle studies. Regardless of the direction, most tongue motions require simultaneous contraction of several tongue muscles to produce hydraulic pressure that alters the functional strength in any untrained tongue movements [10].

Robbins et al. reported that average baseline peak isometric pressure was 41 (36–46) kPa and the pressure increased 7 kPa in older adults after an 8-week program of lingual resistance exercise entailing compression of an air-filled bulb [6]. Van den Steen et al. performed tongue-strengthening exercises for 8 weeks using IOPI in healthy older adults and reported an approximate increase in strength of 26.0 kPa in the anterior maximum isometric pressure (baseline 35.9 ± 6.0 kPa) [14]. Park et al. performed a home-based program for the older adults involving tongue-pressing effortful swallow exercise. Baseline mean tongue pressure was 37.51 ± 15.26 kPa. Four weeks after exercise, the average of the maximum tongue pressure increased by 8.17 kPa [8]. Four weeks of progressive head extension swallowing exercise increased the maximal isometric pressure of 17.6 kPa in this study.

Few studies reported attempts to strengthen the tongue muscles in the form of resistance-swallowing exercise (consistent with exercise principles). Repetitive tongue-holding swallowing exercise was proposed for improving swallowing function in young healthy people, but it showed the same effects as compared to normal swallowing exercise [19]. Park et al. showed that chin-down swallowing exercise improved the lingual strength of healthy young people. However, this exercise was not superior to other tongue-strengthening trainings [20]. The results reinforced our findings in this study.

This study has a few limitations. First, although increasing the degree of head extension requires additional effort during swallowing, evidence is insufficient to show that the resistance increases in proportion to the increasing angle of head extension. However, the maximum head extension angle was increased with exercise. The increment of maximum head extension angle significantly correlated with the increase in the tongue strength, which might support the role of increasing head extension as an appropriate mechanism for achieving overload. Second, we had the participants stare at a point, which was set to maintain the same posture during exercise, but we did not ensure that this direction was perfectly followed in each case. However, we supervised the exercise of all participants to ensure that they followed our instructions correctly. Third, the head extension exercise was conducted with effortful swallows but lingual pressure during swallowing was measured during non-effortful swallows. In terms of training specificity, this limitation might have affected the results of this study.

## 5. Conclusions

Swallowing exercise with progressive head extension increased tongue strength in the older participants. It was easy to monitor the participants anytime and anywhere without any equipment. However, the benefits of this training intervention were not better than other conventional tongue-strengthening exercise. The results suggest that since lingual musculature exhibits atypical response to strength training and all tongue-strength training interventions yield favorable results regardless of the type, it is best to select an exercise option that is easy and most appropriate for the participant and the specific circumstances.

## Figures and Tables

**Figure 1 jcm-10-03419-f001:**
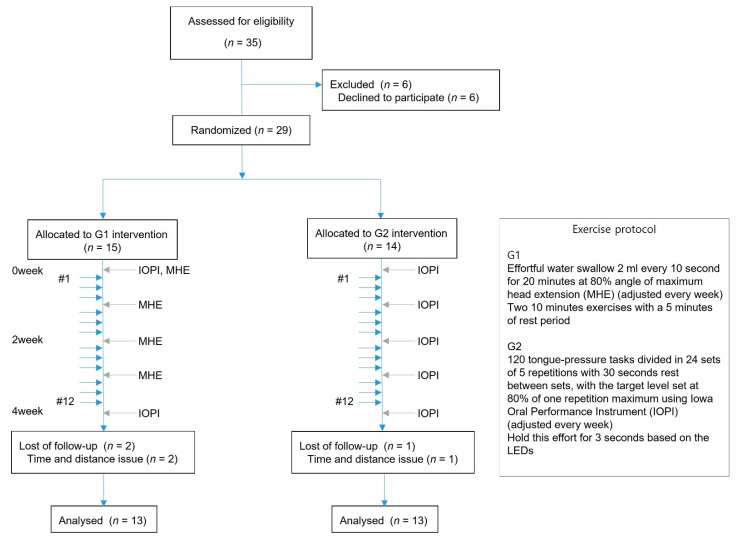
Flow diagram and exercise protocol.

**Figure 2 jcm-10-03419-f002:**
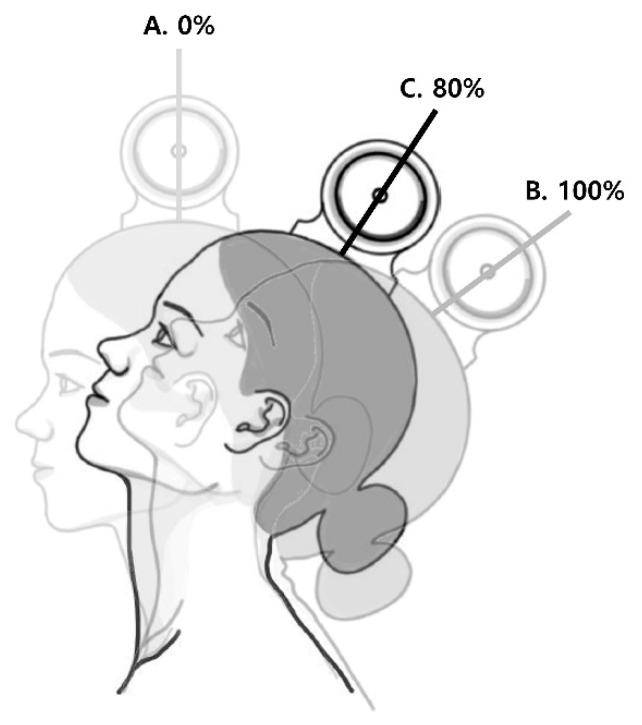
Head extension angle. A. neutral position B. maximal head extension (MHE) C. 80% of MHE.

**Figure 3 jcm-10-03419-f003:**
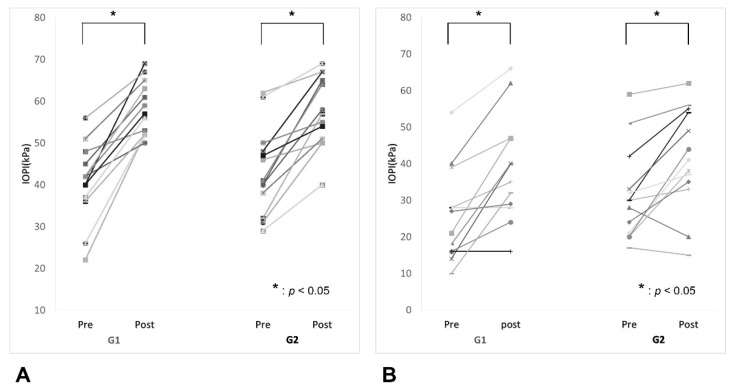
Comparisons of maximal tongue pressure between baseline and post-training sessions in both groups. G1, Tongue progressive resistance exercise group; G2, Tongue isometric exercise group. (**A**) Maximum isometric pressure. Tongue strength was increased significantly in both exercise groups (G1, *p* = 0.000; G2, *p* = 0.000). (**B**) Peak pressure during swallowing. Tongue strength was also increased significantly in both exercise groups (G1, *p* = 0.001; G2, *p* = 0.003).

**Figure 4 jcm-10-03419-f004:**
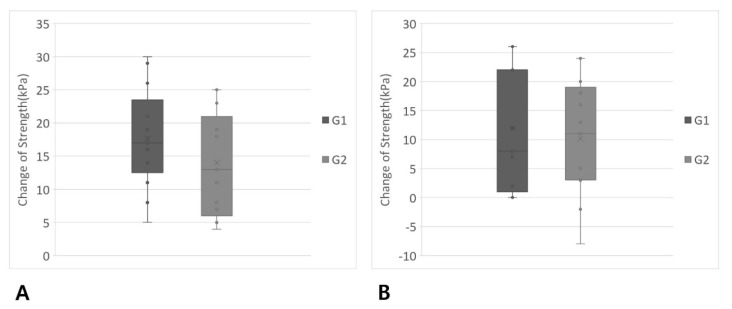
Comparison of the degree of strength increment between the two groups. G1, Tongue progressive resistance exercise group; G2, Tongue isometric exercise group. (**A**) Maximum isometric pressure. There were no significant differences between the groups (G1, 17.6 ± 7.5 kPa and G2, 14.0 ± 7.9 kPa, *p* = 0.244). (**B**) Peak pressure during swallowing. No significant differences were detected between groups. (G1, 11.9 ± 10.3 kPa and G2, 10.2 ± 10.1 kPa, *p* = 0.662). Box: 1st quartile and 3rd quartile; Whisker: minimum and maximum; Line: median; X: average.

**Table 1 jcm-10-03419-t001:** General characteristics of participants in this study.

	Tongue Progressive Resistance Exercise; G1 (*n* = 13)	Tongue Isometric Exercise; G2 (*n* = 13)	*p*-Value
Age (years)	72.7 ± 7.3 (65–87)	73.2 ± 5.7 (65–82)	0.835
Sex			
Male	4	1	0.135
Female	9	12	
Mini-mental status exam	28.6 ± 1.3 (26–30)	28.2 ± 1.3 (26–30)	0.387
Baseline maximum head extension angle (degrees)	39.6 ± 9.9 (25–55)		
4th week maximum head extension angle (degrees)	57.7 ± 7.8 (40–70)		
Baseline maximum isometric pressure (kPa)	40.5 ± 9.2 (22–56)	43.5 ± 10.4 (29–62)	0.455
Baseline peak pressure during swallowing (kPa)	26.1 ± 12.4 (10–54)	31.3 ± 12.6 (17–59)	0.297

## Data Availability

Data presented in this study are provided by the corresponding authors upon reasonable request.

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
