# Peer review of "Effect of Progressive Head Extension Swallowing Exercise on Lingual Strength in the Elderly: A Randomized Controlled Trial"

_jcm, 2021, doi:10.3390/jcm10153419_

Round 1
Reviewer 1 Report
Thank you for this work. I thought that your manuscript was very clearly written and accessible to the readership. The context was well described and sufficiently critical and you made a good case for the justification of your study. I found you methodology easy to follow and had no reason to think that your study could not be replicated following your detailed descriptions.
I would like a description of how you managed to "blind" your assessments please. Results were presented coherently and with a good use of figures.
Your discussion was appropriate and stimulating. Congratulations.
Author Response
Thank you for this work. I thought that your manuscript was very clearly written and accessible to the readership. The context was well described and sufficiently critical and you made a good case for the justification of your study. I found you methodology easy to follow and had no reason to think that your study could not be replicated following your detailed descriptions.
I would like a description of how you managed to "blind" your assessments please.
One independent assessor measured tongue pressure in all participants. He did not know which group the participant belonged to, and the results were kept blind by having him record the results on a coded paper sheet after the measurements. The statistical analyst was also allowed to analyze without knowing which treatment the two groups received.
Results were presented coherently and with a good use of figures.
Your discussion was appropriate and stimulating. Congratulations.
Thank you very much.
Reviewer 2 Report
In this manuscript (jcm-1317624), the authors have compared two types of tongue-strengthening exercises to improve lingual pressure. They showed no significant difference between the exercises and demonstrated that they have equal efficacy.
I have some concerns about this MS:
Major comments
1.Methods
Some readers may have difficulties imagining the 80% maximum head extension. I would like to propose adding a figure to show the posture.
2.Discussion
In addition, it is better to present average values of tongue pressure in same-aged healthy people measured by IOPI in previous studies as a reference and compare them with this study’s data.
Minor comments
There are two Figure 2. Isn’t the latter one Figure 3?
Author Response
Reviewer 2
In this manuscript (jcm-1317624), the authors have compared two types of tongue-strengthening exercises to improve lingual pressure. They showed no significant difference between the exercises and demonstrated that they have equal efficacy.
I have some concerns about this MS:
Major comments
1.Methods
Some readers may have difficulties imagining the 80% maximum head extension. I would like to propose adding a figure to show the posture.
We added the figure showing the head posture.
2.Discussion
In addition, it is better to present average values of tongue pressure in same-aged healthy people measured by IOPI in previous studies as a reference and compare them with this study’s data.
We added the average values of tongue pressure in older adults as you recommended.
Minor comments
There are two Figure 2. Isn’t the latter one Figure 3?
We corrected the figure numbers. Thank you so much.